# WHICH TASKS SHOULD BE LEARNED TOGETHER IN MULTI-TASK LEARNING?

## ABSTRACT

Many computer vision applications require solving multiple tasks in real-time. A neural network can be trained to solve multiple tasks simultaneously using *multi-task learning*. This saves computation at inference time as only a single network needs to be evaluated. Unfortunately, this often leads to inferior overall performance as task objectives can compete, which consequently poses the question: **which tasks should and should not be learned together in one network when employing multi-task learning?** We systematically study task cooperation and competition and propose a framework for assigning tasks to a few neural networks such that cooperating tasks are computed by the same neural network, while competing tasks are computed by different networks. Our framework offers a time-accuracy trade-off and can produce better accuracy using less inference time than not only a single large multi-task neural network but also many single-task networks.

## 1 INTRODUCTION

Many applications, especially robotics and autonomous vehicles, are chiefly interested in using multi-task learning to reduce the inference time and computational complexity required to estimate many characteristics of visual input. For example, an autonomous vehicle may need to detect the location of pedestrians, determine a per-pixel depth, and predict objects' trajectories, all within tens of milliseconds. In multi-task learning, multiple tasks are solved at the same time, typically with a single neural network. In addition to reduced inference time, solving a set of tasks jointly rather than independently can, in theory, have other benefits such as improved prediction accuracy, increased data efficiency, and reduced training time.

Unfortunately, the quality of predictions are often observed to suffer when a network is tasked with making multiple predictions. This is because learning objectives can have complex and unknown dynamics and may compete. In fact, multi-task performance can suffer so much that smaller independent networks are often superior (as we will see in the experiments section). We refer to any situation in which the competing priorities of the network cause poor task performance as *crosstalk*.

On the other hand, when task objectives do not interfere much with each other, performance on both tasks can be maintained or even improved when jointly trained. Intuitively, this loss or gain of quality seems to depend on *the relationship between the jointly trained tasks*.

Prior work has studied the relationship between tasks for transfer learning (Zamir et al. (2018)). However, we find that transfer relationships are not highly predictive of multi-task relationships. In addition to studying multi-task relationships, we attempt to determine how to produce good prediction accuracy under a limited inference time budget by assigning competing tasks to separate networks and cooperating tasks to the same network.

More concretely, this leads to the following problem: Given a set of tasks, $\mathcal{T}$, and a computational budget $b$ (e.g., maximum allowable inference time), what is the optimal way to assign tasks to networks with combined cost $\leq b$ such that a combined measure of task performances is maximized?

To this end, we develop a computational framework for choosing the best tasks to group together in order to have a small number of separate deep neural networks that completely cover the task set and that maximize task performance under a given computational budget. We make the intriguing

Figure 1: **Given five tasks to solve, there are many ways that they can be split into task groups for multi-task learning. How do we find the best one?** We propose a computational framework that, for instance, suggests the following grouping to achieve the lowest total loss, using a computational budget of 2.5 units: train network A to solve Semantic Segmentation, Depth Estimation, and Surface Normal Prediction; train network B to solve Keypoint Detection, Edge Detection, and Surface Normal Prediction; train network C with a less computationally expensive encoder to solve Surface Normal Prediction alone; including Surface Normals as an output in the first two networks were found advantageous for improving the other outputs, while the best Normals were predicted by the third network. This task grouping outperforms all other feasible ones, including learning all five tasks in one large network or using five dedicated smaller networks.

observation that the inclusion of an additional task in a network can potentially improve the accuracy of the other tasks, even though the performance of the added task might be poor. This can be viewed as *regularizing* or *guiding* the loss of one task by adding an additional loss, as often employed in curriculum learning or network regularization Bengio et al. (2009). Achieving this, of course, depends on picking the proper regularizing task – our system can take advantage of this phenomenon, as schematically shown in Figure 1.

This paper has two main contributions. In Section 3, we outline a framework for systematically assigning tasks to networks in order to achieve the best total prediction accuracy with a limited inference-time budget. We then analyze the resulting accuracy and show that selecting the best assignment of tasks to groups is critical for good performance. Secondly, in Section 6, we analyze situations in which multi-task learning helps and when it doesn't, quantify the compatibilities of various task combinations for multi-task learning, compare them to the transfer learning task affinities, and discuss the implications. Moreover, we analyze the factors that influence multi-task affinities.

## 2 PRIOR WORK

**Multi-Task Learning:** See Ruder (2017) for a good overview of multi-task learning. The authors identify two clusters of contemporary techniques that we believe cover the space well, hard parameter sharing and soft parameter sharing. In brief, the primary difference between the majority of the existing works and our study is that we wish to understand the relationships between tasks and find compatible groupings of tasks for any given set of tasks, rather than designing a neural network architecture to solve a particular fixed set of tasks well.

A known contemporary example of hard parameter sharing in computer vision is UberNet (Kokkinos (2017)). The authors tackle 7 computer vision problems using hard parameter sharing. The authors focus on reducing the computational cost of training for hard parameter sharing, but experience a rapid degradation in performance as more tasks are added to the network. Hard parameter sharing is also used in many other works such as (Thrun (1996); Caruana (1997); Nekrasov et al. (2018); Dvornik et al. (2017); Kendall et al. (2018); Bilen & Vedaldi (2016); Pentina & Lampert (2017); Doersch & Zisserman (2017); Zamir et al. (2016); Long et al. (2017); Mercier et al. (2018); d. Miranda et al. (2012); Zhou et al. (2018); Rudd et al. (2016)).

Other works, such as (Sener & Koltun (2018)) and (Chen et al. (2018b)), aim to dynamically reweight each task's loss during training. The former work finds weights that provably lead to a Pareto-optimal solution, while the latter attempts to find weights that balance the influence of each task on network weights. Finally, (Bingel & Søgaard (2017)) studies task interaction for NLP.

In soft or partial parameter sharing, either there is a separate set of parameters per task, or a significant fraction of the parameters are unshared. The models are tied together either by information sharing or by requiring parameters to be similar. Examples include (Dai et al. (2016); Duong et al. (2015); Misra et al. (2016); Tessler et al. (2017); Yang & Hospedales (2017); Lu et al. (2017)).

The canonical example of soft parameter sharing can be seen in (Duong et al. (2015)). The authors are interested in designing a deep dependency parser for languages such as Irish that do not have much treebank data available. They tie the weights of two networks together by adding an L2 distance penalty between corresponding weights and show substantial improvement.

Another example of soft parameter sharing is Cross-stitch Networks (Misra et al. (2016)). Starting with separate networks for two tasks, the authors add 'cross-stitch units' between them, which allow each network to peek at the other network's hidden layers. This approach reduces but does not eliminate task interfearence, and the overall performance is less sensitive to the relative loss weights.

Unlike our method, none of the aforementioned works attempt to discover good groups of tasks to train together. Also, soft parameter sharing does not reduce inference time, a major goal of ours.

**Transfer Learning:** Transfer learning (Pratt (1993); Helleputte & Dupont (2009); Silver & Bennett (2008); Finn et al. (2016); Mihalkova et al. (2007); Niculescu-Mizil & Caruana (2007); Luo et al. (2017); Razavian et al. (2014); Pan & Yang (2010); Mallya & Lazebnik (2018); Fernando et al. (2017); Rusu et al. (2016)) is similar to multi-task learning in that solutions are learned for multiple tasks. Unlike multi-task learning, however, transfer learning methods often assume that a model for a source task is given and then adapt that model to a target task. Transfer learning methods generally neither seek any benefit for source tasks nor a reduction in inference time as their main objective.

**Neural Architecture Search (NAS):** Many recent works search the space of deep learning architectures to find ones that perform well (Zoph & Le, 2017; Liu et al., 2018; Pham et al., 2018; Xie et al., 2019; Elsken et al., 2019; Zhou et al., 2019; Baker et al., 2017; Real et al., 2018). This is related to our work as we search the space of task groupings. Just as with NAS, the found task groupings often perform better than human-engineered ones.

**Task Relationships:** Our work is most related to *Taskonomy* (Zamir et al. (2018)), where the authors studied the relationships between visual tasks for *transfer learning* and introduced a dataset with over 4 million images and corresponding labels for 26 tasks. This was followed by a number of recent works, which further analyzed task relationships (Pal & Balasubramanian (2019); Dwivedi & Roig. (2019); Achille et al. (2019); Wang et al. (2019)) for transfer learning. While they extract relationships between these tasks for *transfer learning*, we are interested in the *multi-task learning* setting. Interestingly, we find notable differences between transfer task affinity and multi-task affinity. Their method also differs in that they are interested in labeled-data efficiency and not inference-time efficiency. Finally, the transfer quantification approach taken by Taskonomy (readout functions) is only capable of finding relationships between the high-level bottleneck representations developed for each task, whereas structural similarities between tasks at all levels are potentially relevant for multi-task learning.

## 3 TASK GROUPING FRAMEWORK

Our goal is to find an assignment of tasks to networks that results in the best overall loss. Our strategy is to select from a large set of candidate networks to include in our final solution.

We define the problem as follows: We want to minimize the overall loss on a set of tasks $\mathcal{T} = \{t_1, t_2, ..., t_k\}$ given a limited inference time budget, $b$, which is the total amount of time we have to complete all tasks. Each neural network that solves some subset of $\mathcal{T}$ and that could potentially be a part of the final solution is denoted by $n$. It has an associated inference time cost, $c_n$, and a loss for each task, $\mathcal{L}(n, t_i)$ (which is $\infty$ for each task the network does not attempt to solve). A solution $\boldsymbol{S}$ is a set of networks that together solve all tasks. The computational cost of a solution is $\text{cost}(\boldsymbol{S}) = \sum_{n \in \boldsymbol{S}} c_n$. The loss of a solution on a task, $\mathcal{L}(\boldsymbol{S}, t_i)$, is the lowest loss on that task among the solution's networks[1], $\mathcal{L}(\boldsymbol{S}, t_i) = \min_{n \in \boldsymbol{S}} \mathcal{L}(n, t_i)$. The overall performance for a solution is $\mathcal{L}(\boldsymbol{S}) = \sum_{t_i \in \mathcal{T}} \mathcal{L}(\boldsymbol{S}, t_i)$.

We want to find the solution with the lowest overall loss and a cost that is under our budget, $\boldsymbol{S}_b = \text{argmin}_{\boldsymbol{S}:\text{cost}(\boldsymbol{S}) \leq b} \mathcal{L}(\boldsymbol{S})$.

---

[1]In principle, it may be possible to create an even better-performing ensemble when multiple networks solve the same task, though we do not explore this.

### 3.1 WHICH CANDIDATE NETWORKS TO CONSIDER?

For a given task set $\mathcal{T}$, we wish to determine not just how well each *pair* of tasks performs when trained together, but also how well each *combination* of tasks performs together so that we can capture higher-order task relationships. To that end, our candidate set of networks contains all $2^{|\mathcal{T}|} - 1$ possible groupings: $\binom{|\mathcal{T}|}{1}$ networks with one task, $\binom{|\mathcal{T}|}{2}$ networks with two tasks, $\binom{|\mathcal{T}|}{3}$ networks with three tasks, etc. For the five tasks we use in our experiments, this is 31 networks, of which five are single-task networks.

The size of the networks is another design choice, and to somewhat explore its effects we also include 5 single task networks each with half of the computational cost of a standard network. This brings our total up to 36 networks.

### 3.2 NETWORK SELECTION

Consider the situation in which we have an initial candidate set $\boldsymbol{C}_0 = \{n_1, n_2, ..., n_m\}$ of fully-trained networks that each solve some subset of our task set $\mathcal{T}$. Our goal is to choose a subset of $\boldsymbol{C}_0$ that solve all the tasks with total inference time under budget $b$ and the lowest overall loss. More formally, we want to find a solution $\boldsymbol{S}_b = \mathrm{argmin}_{\boldsymbol{S} \subseteq \boldsymbol{C}_0 : \mathrm{cost}(\boldsymbol{S}) \leq b} \mathcal{L}(\boldsymbol{S})$.

It can be shown that solving this problem is NP-hard in general (reduction from SET-COVER). However, many techniques exist that can optimally solve *most* reasonably-sized instances of problems like these in acceptable amounts of time. All of these techniques produce the same solutions. We chose to use a branch-and-bound-like algorithm for finding our optimal solutions (shown as Algorithm 1 in the Appendix), but in principle the exact same solutions could be achieved by other optimization methods, such as encoding the problem as a binary integer program (BIP) and solving it in a way similar to Taskonomy (Zamir et al. (2018)).

Most contemporary MTL works use fewer than 4 unique task types, but in principal, the NP-hard nature of the optimization problem does limit the number of candidate solutions that can be considered. However, using synthetic inputs, we found that our branch-and-bound like approach requires less time than network training for all $2^{|\mathcal{T}|} - 1 + |\mathcal{T}|$ candidates for fewer than ten tasks. Scaling beyond that would require approximations or stronger optimization techniques.

### 3.3 APPROXIMATIONS FOR REDUCING TRAINING TIME COMPLEXITY

This section describes two techniques for reducing the training time required to obtain a collection of networks as input to the network selection algorithm. Our goal is to produce task groupings with results similar to the ones produced by the complete search, but with less training time burden. Both techniques involve predicting the performance of a network without actually training it to convergence. The first technique involves training each of the networks for a short amount of time, and the second involves inferring how networks trained on more than two tasks will perform based on how networks trained on two tasks perform.

#### 3.3.1 EARLY STOPPING PRIOR TO CONVERGENCE

We found a moderately high correlation (Pearson's $r = 0.49$) between the validation loss of our neural networks after a pass through just 20% of our data and the final test loss of the fully trained networks. This implies that the task relationship trends stabilize early. We fine that we can get decent results by running network selection on the lightly trained networks, and then simply training the chosen networks to convergence.

For our setup, this technique reduces the training time burden by about **20x** over fully training all candidate networks and would require fewer than 150 GPU hours to execute. This is only 35% training-time overhead. Obviously, this technique does come with a prediction accuracy penalty. Because the correlation between early network performance and final network performance is not perfect, the decisions made by network selection are no longer guaranteed to be optimal once networks are trained to convergence. We call this approximation the Early Stopping Approximation (ESA) and present the results of using this technique in Section 5.

### 3.3.2 PREDICT HIGHER-ORDER FROM LOWER-ORDER

Do the performances of a network trained with tasks $A$ and $B$, another trained with tasks $A$ and $C$, and a third trained with tasks $B$ and $C$ tell us anything about the performance of a network trained on tasks $A$, $B$, and $C$? As it turns out, the answer is yes. Although this ignores complex task interactions and nonlinearities, a simple average of the first-order networks' accuracies was a good indicator of the accuracy of a higher-order network. Experimentally, this prediction strategy has an average max ratio error of only 5.2% on our candidate networks.

Using this strategy, we can predict the performance of all networks with three or more tasks using the performance of all of the fully trained two task networks. First, simply train all networks with two or fewer tasks to convergence. Then predict the performance of higher-order networks. Finally, run network selection on both groups.

With our setup (see Section 4), this strategy saves training time by only about 50%, compared with 95% for the early stopping approximation, and it still comes with a prediction quality penalty. However, this technique requires only a quadratic number of networks to be trained rather than an exponential number, and would therefore win out when the number of tasks is large.

We call this strategy the Higher Order Approximation (HOA), and present its results in Section 5.

## 4 EXPERIMENTAL SETUP

We perform our evaluation using the Taskonomy dataset (Zamir et al. (2018)), which is currently the largest multi-task dataset in vision with diverse tasks. The data was obtained from 3D scans of about 600 buildings. There are 4,076,375 examples, which we divided into 3,974,199 training instances, 52,000 validation instances, and 50,176 test instances. There was no overlap in the buildings that appeared in the training and test sets. All data labels were normalized ($\bar{x} = 0, \sigma = 1$).

Our framework is agnostic to the particular set of tasks. We have chosen to perform the study using five tasks in Taskonomy: *Semantic Segmentation, Depth Estimation, Surface Normal Prediction, Keypoint Detection*, and *Edge Detection*, so that one semantic task, two 3D tasks, and two 2D tasks are included. These tasks were chosen to be representative of major task categories, but also to have enough overlap in order to test the hypothesis that similar tasks will train well together. Cross-entropy loss was used for Semantic Segmentation, while an $L1$ loss was used for all other tasks.

**Network Architecture:** The proposed framework can work with any network architecture. In our experiments, all of the networks used a standard encoder-decoder architecture with a modified Xception (Chollet (2017)) encoder. Our choice of architecture is not critical and was chosen for reasonably fast inference time performance. The Xception network encoder was simplified to have 17 layers and the middle flow layers were reduced to having 512 rather than 728 channels. All max-pooling layers were replaced by $2 \times 2$ convolution layers with a stride of 2 (similar to Chen et al. (2018a)). The full-size encoder had about 4 million parameters. All networks had an input image size of 256x256. We measure inference time in units of the time taken to do inference for one of our full-size encoders. We call this a *Standard Network Time (SNT)*. This corresponds to 2.28 billion multiply-adds and about 4 ms/image on a single Nvidia RTX 2080 Ti.

Our decoders were designed to be lightweight and have four transposed convolutional layers (Noh et al. (2015)) and four separable convolutional layers (Chollet (2017)). Every decoder has about 116,000 parameters. All training was done using PyTorch (Paszke et al. (2017)) with Apex for fp16 acceleration (Micikevicius et al. (2017)).

**Trained Networks:** As described in Section 3.1, we trained 31 networks with full sized encoders and standard decoders. 26 were multi-task networks and 5 were single task networks. Another five single-task networks were trained, each having a half-size encoder and a standard decoder. These 36 networks were included in network optimization as $C_0$. 20 smaller, single-task networks of various sizes were also trained to be used in the baselines and the analysis of Section 6, but not used for network selection. In order to produce our smaller models, we shrunk the number of channels in every layer of the encoder such that it had the appropriate number of parameters and flops.

The training loss we used was the unweighted mean of the losses for the included tasks. Networks were trained with an initial learning rate of 0.2, which was reduced by half every time the training

loss stopped decreasing. Networks were trained until their validation loss stopped improving, typically requiring only 4-8 passes through the dataset. The network with the highest validation loss (checked after each epoch of 20% of our data) was saved.

The performance scores used for network selection were calculated on the validation set. We computed solutions for inference time budgets from 1 to 5 at increments of 0.5. Each solution chosen was evaluated on the test set.

## 4.1 BASELINES

We compare our results with conventional methods, such as five single-task networks and a single network with all tasks trained jointly.

We also compare with two multi-task methods in the literature. The first one is Sener & Koltun (2018). We found that their algorithm under-weighted the Semantic Segmentation task too aggressively, leading to poor performance on the task and poor performance overall compared to a simple sum of task losses. We speculate that this is because semantic segmentation's loss behaves differently from the other losses. Next we compared to GradNorm (Chen et al. (2018b)). The results here were also slightly worse than classical MTL with uniform task weights. In any event, these techniques are orthogonal to ours and can be used in conjunction for situations in which they lead to better solutions than simply summing losses.

Finally, we compare our results to two control baselines illustrative of the importance of making good choices about which tasks to train together, 'Random' and 'Pessimal.' 'Random' is a solution consisting of valid random task groupings that solve our five tasks. The reported values are the average of a thousand random trials. 'Pessimal' is a solution in which we choose the networks that lead to the worst overall performance, though the solution's performance on each task is still the best among its networks.

Each baseline was evaluated with multiple encoder sizes so that all models' results could be compared at many inference time budgets.

## 5 TASK GROUPING EVALUATION

Figure 2: **The task groups picked by each of our techniques for integer budgets between 1 and 5.** Networks are shown as ◯ (full-size) or ◦ (half-size). Networks are connected to the tasks for which they compute predictions. *s: Semantic Segmentation, d: Depth Estimation, n: Surface Normal Prediction, k: Keypoint Detection, e: Edge Detection*. Dotted edges represent unused decoders. For example, the highlighted solution consists of two half-size networks and a full-size network. The full-size network solves Depth Estimation, Surface Normal Prediction, and Keypoint Detection. One half-size network solves Semantic Segmentation and the other solves Edge Detection. The total loss for all five tasks is 0.455. The groupings for fractional budgets are shown in the appendix.

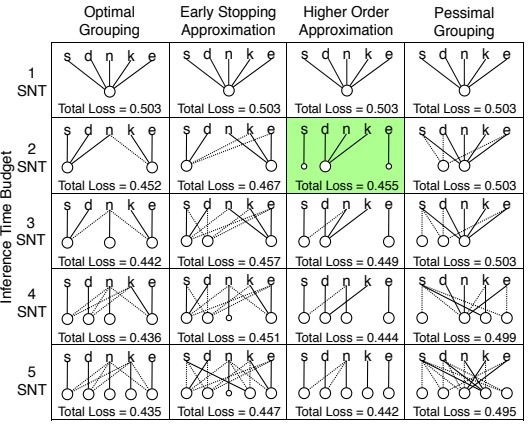

Figure 2 shows the task groups that were chosen for each technique, and Figure 3 shows the performance of these groups along with those of our baselines. We can see that each of our methods outperforms our traditional baselines for every computational budget.

When the computational budget is only 1 SNT, all of our methods must select the same model—a traditional multi-task network with a 1 SNT encoder and five decoders. This strategy outperforms GradNorm, Sener & Koltun (2018), and individual training. However, solutions that utilize multiple networks outperform this traditional strategy for every budget > 1.5—better performance can always be achieved by grouping tasks according to their compatibility.

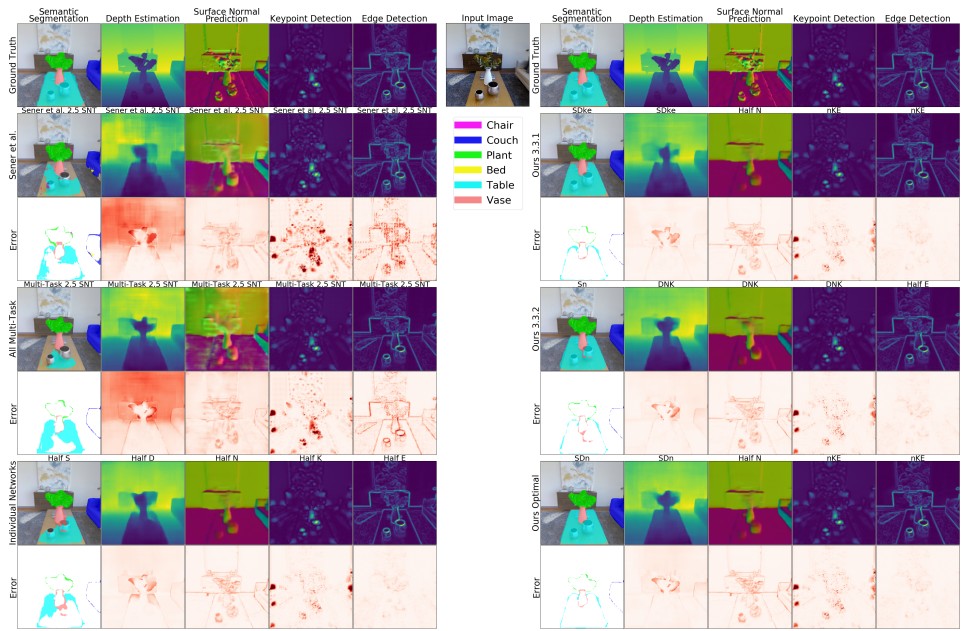

Figure 4: Qualitative results for our baselines (left) and our techniques (right). All solutions allowed 2.5 SNT.

When the computational budget is effectively unlimited (5 SNT), our optimal method picks five networks, each of which is used to make predictions for a separate task. However, three of the networks are trained with three tasks each, while only two are trained with one task each. This shows that the representations learned through multi-task learning were found to be best for three of our tasks (s, d, and e), whereas two of our tasks (n and k) are best solved individually.

We also see that our optimal technique using 2.5 SNT and our Higher Order Approximation using 3.5 SNT can both outperform five individual networks (which uses 5 SNT).

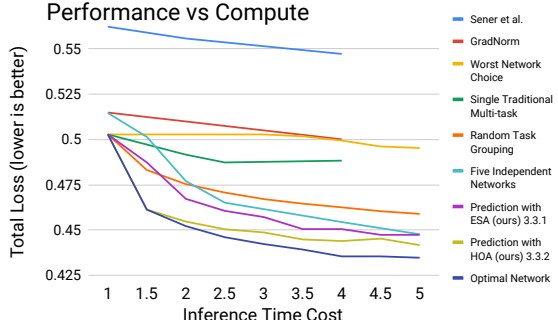

Figure 3: The performance/inference time trade-off for various methods. Data presented tabularly in Table 7.

In order to determine how these task groupings generalize to other architectures, we retrained our best solution for 3 SNT using resnet18 (He et al. (2016)). The results in Table 1 suggest that good task groupings for one architecture are likely to be good in another, though to a lesser extent. Task affinities seem to be somewhat architecture-dependent, so for the very best results, task selection must be run for each architecture choice.

| resnet18 | Total Loss |
|---|---|
| All-in-one (triple-size resnet18) | 0.50925 |
| Five Individual (resnet18s .6-size each) | 0.53484 |
| **nKE, SDn, N** (3 standard resnet18's) | **0.50658** |

Table 1: The performance of our best 3 SNT solution found using Xception but evaluated on ResNet18.

Figure 4 allows qualitative comparison between our methods and our baselines. We can see clear visual issues with each of our baselines that are not present in our methods. Both of our approximate methods produce predictions similar to the optimal task grouping.

## 6 ANALYSES OF TASK RELATIONSHIPS

The data generated by the above evaluation presents an opportunity to analyze how tasks interact in a multi-task setting, and allows us to compare with some of the vast body of research in transfer learning, such as Taskonomy (Zamir et al. (2018)).

| | | | | Relative Performance On | | | |
| | | SemSeg | Depth | Normals | Keypoints | Edges | Average |
|---|---|---|---|---|---|---|---|
| Trained With | SemSeg | – | -5.41% | -11.29% | -4.32% | -34.64% | -13.92% |
| | Depth | 4.17% | – | -3.55% | 3.49% | 3.76% | 1.97% |
| | Normals | 8.50% | 2.48% | – | 1.37% | 12.33% | 6.17% |
| | Keypoints | 4.82% | 1.38% | -0.02% | – | -5.26% | 0.23% |
| | Edges | 3.07% | -0.92% | -4.42% | 1.37% | – | -0.23% |
| | Average | 5.14% | -0.62% | -4.82% | 0.48% | -5.95% | -1.15% |

Table 2: **The first-order multi-task learning relationships between tasks.** The table lists the performance of every task when trained as a pair with every other task. For instance, when Depth is trained with SemSeg, SemSeg performs 4.17% better than when SemSeg is trained alone on a half-size network.

| | Depth | Normals | Keypoints | Edges |
|---|---|---|---|---|
| SemSeg | -0.62% | -1.39% | 0.25% | -15.78% |
| Depth | | -0.54% | 2.43% | 1.42% |
| Normals | | | 0.67% | 3.95% |
| Keypoints | | | | -1.95% |

Table 3: **The *multi-task* learning affinity between pairs of tasks.** These values show the average change in the performance of two tasks when trained as a pair, relative to when they are trained separately.

| | Depth | Normals | Keypoints | Edges |
|---|---|---|---|---|
| SemSeg | 1.740 | 1.828 | 0.723 | 0.700 |
| Depth | | 1.915 | 0.406 | 0.468 |
| Normals | | | 0.089 | 0.118 |
| Keypoints | | | | 0.232 |

Table 4: **The *transfer* learning affinities** between pairs of tasks according to the authors of Taskonomy (Zamir et al. (2018)). Forward and backward transfer affinities are averaged.

In order to determine the between task affinity for multi-task learning, we took the average of our first-order relationships matrix (Table 2) and its transpose. The result is shown in Table 3. The pair with the highest affinity by this metric are Surface Normal Prediction and 2D Edge Detection. Our two 3D tasks, Depth Estimation and Surface Normal Prediction, do not score highly on this similarity metric. This contrasts with the findings for transfer learning in Taskonomy (Table 4), in which they have the highest affinity. Our two 2D tasks also do not score highly. We speculate that the Normals task naturally preserves edges, while Depth and Normals (for example) don't add much training signal to each other. See Section A.3 for more on factors that influence multi-task affinity.

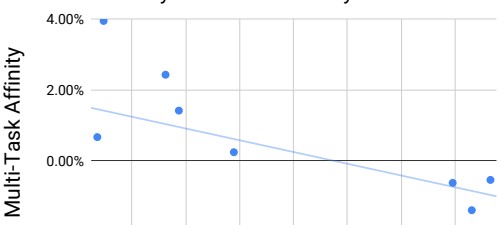

Figure 5: Task affinities for multi-task learning vs. transfer learning. The correlation (Pearson's $r$) is $-0.54$, $p = 0.13$. One outlier is removed.

Figure 5 depicts the relationship between transfer learning affinities and multi-task affinities, which surprisingly seem to be negatively correlated in our high-data scenario. This suggests that it might be better to train dissimilar tasks together. This could be because dissimilar tasks are able to provide stronger and more meaningful regularization. More research is necessary to discover when and if this correlation and explanation hold.

# 7 CONCLUSION

We describe the problem of task compatibility as it pertains to multi-task learning. We provide an algorithm and computational framework for determining which tasks should be trained jointly and which tasks should be trained separately. Our solution can take advantage of situations in which joint training is beneficial to some tasks but not others in the same group. For many use cases, this framework is sufficient, but it can be costly at training time. Hence, we offer two strategies for coping with this issue and evaluate their performance. Our methods outperform single-task networks, a multi-task network with all tasks trained jointly, as well as other baselines. Finally, we use this opportunity to analyze how particular tasks interact in a multi-task setting and compare that with previous results on transfer learning task interactions.

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

## A  APPENDIX

### A.1  NETWORK SELECTION ALGORITHM

---
**Algorithm 1** Get Best Networks

---
**Input:** $C_r$, a running set of candidate networks, each with an associated cost $c \in \mathbb{R}$ and a performance score for each task the network solves. Initially, $C_r = C_0$
**Input:** $S_r \subseteq C_0$, a running solution, initially Ø
**Input:** $b_r \in \mathbb{R}$, the remaining time budget, initially $b$

1:  **function** GETBESTNETWORKS($C_r, S_r, b_r$)
2:      $C_r \leftarrow$ FILTER($C_r, S_r, b_r$)
3:      $C_r \leftarrow$ SORT($C_r$)                              ▷ Most promising networks first
4:      $Best \leftarrow S_r$
5:      **for** $n \in C_r$ **do**
6:          $C_r \leftarrow C_r \setminus n$                    ▷ \ is set subtraction.
7:          $S_i \leftarrow S_r \cup \{n\}$
8:          $b_i \leftarrow b_r - c_n$
9:          $Child \leftarrow$ GETBESTNETWORKS($C_r, S_i, b_i$)
10:          $Best \leftarrow$ BETTER($Best, Child$)
11:      **return** $Best$

12: **function** FILTER($C_r, S_r, b_r$)
13:      Remove networks from $C_r$ with $c_n > b_r$.
14:      Remove networks from $C_r$ that cannot improve $S_r$'s performance on any task.
15:      **return** $C_r$

16: **function** BETTER($S_1, S_2$)
17:      **if** $C(S_1) < C(S_2)$ **then**
18:          **return** $S_1$
19:      **else**
20:          **return** $S_2$

---

Algorithm 1 chooses the best subset of networks in our collection, subject to the inference time budget constraint. The algorithm recursively explores the space of solutions and prunes branches that cannot lead to optimal solutions. The recursion terminates when the budget is exhausted, at which point $C_r$ becomes empty and the loop body does not execute.

The sorting step on line 3 requires a heuristic upon which to sort. We found that ranking models based on how much they improve the current solution, $S$, works well. It should be noted that this algorithm always produces an optimal solution, regardless of which sorting heuristic is used. However, better sorting heuristics reduce the running time because subsequent iterations will more readily detect and prune portions of the search space that cannot contain an optimal solution. In our setup, we tried variants of problems with 5 tasks and 36 networks, and all of them took less than a second to solve.

The definition of the BETTER() function is application-specific. For our experiments, we prefer networks that have the lowest total loss across all five tasks. Other applications may have hard

performance requirements for some of the tasks, and performance on one of these tasks cannot be sacrificed in order to achieve better performance on another task. Such application-specific constraints can be encoded in BETTER().

## A.2 ANALYSIS ON SMALLER TASK SETS

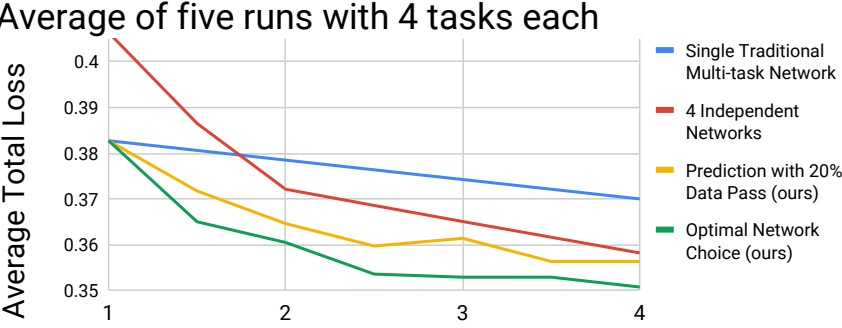

Figure 6: Our experiments re-run on all 4-task subsets, then averaged.

In order to determine how well network selection works for different task sets, we re-ran network selection on all five 4-task subsets of our task set. The performance average of all 5 sets is shown in Figure 6. We see that our techniques generalize at least to subsets of our studied tasks.

## A.3 DEPTH AND NORMALS IN A MORE TYPICAL SETTING

The finding that Depth and Normals don't cooperate is counter to much of the multi-task learning literature such as Wang et al. (2016), Qi et al. (2018), and Zhang et al. (2019). However, the majority of these works use training sets with fewer than 100k instances, while we use nearly 4 million training instances. Table 5 shows the loss obtained on our setup when we limit to only 100k training instances. The fact that task affinities can change depending on the amount of available training data demonstrates the necessity of using an empirical approach like ours for finding task affinities and groupings.

| 100k training instances | Depth Test Loss | Normals Test Loss |
|---|---|---|
| Depth Alone | 0.265 | - |
| Normals Alone | - | 0.1398 |
| Joint Depth + Normals | **0.2525** | **0.1319** |

Table 5: Positive task affinity between depth and normals in a low data setting.

## A.4 TABULAR DATA

|  | Ours Optimal | Single 20% pass 3.3.1 | Higher Order 3.3.2 |
|---|---|---|---|
| 1 | SDNKE | SDNKE | SDNKE |
| 1.5 | DNKE, S | SDNK, E | DNKE, S |
| 2 | nKE, SDN | SDke, NKE | DNK, E, S |
| 2.5 | nKE, SDn, N | SDke, nKE, N | DNK, E, Sn |
| 3 | nKE, SDn, N | SDne, sdke, NKE | DNK, E, Sn |
| 3.5 | nKE, Snk, Dnk, N | SDne, sdke, nKE, N | DnK, E, Sn, N |
| 4 | nKE, Snk, Dnk, N | SDne, sdke, nKE, N | Sn, DK, E, N |
| 4.5 | nKE, Snk, Dnk, N | sDne, sdke, nKE, N, Snk | Sn, E, K, Dn, N |
| 5 | nkE, Snk, Dnk, N, K | sDne, sdke, nKE, N, Snk | Sn, E, K, Dn, N |

Table 6: **The task groups picked by each of our techniques for every budget choice between 1 and 5.** Networks are shown as a list of letters corresponding to each task the network contains. *S: Semantic Segmentation, D: Depth Estimation, N: Surface Normal Prediction, K: Keypoint Detection, E: Edge Detection.* Capital letters denote that a solution used that network's prediction for that task. Half-sized networks are shown in red.

| Time Budget | 1 | 1.5 | 2 | 2.5 | 3 | 3.5 | 4 | 4.5 | 5 |
|---|---|---|---|---|---|---|---|---|---|
| Sener et al. | 0.562 | | 0.556 | 0.551 | | | 0.547 | | |
| GradNorm | 0.515 | | | | | | 0.500 | | |
| Pessimal Grouping | **0.503** | 0.503 | 0.503 | 0.503 | 0.503 | 0.502 | 0.499 | 0.496 | 0.495 |
| Traditional MTL | **0.503** | | 0.492 | 0.487 | | | 0.488 | | |
| Random Groupings | **0.503** | 0.483 | 0.475 | 0.471 | 0.467 | 0.464 | 0.462 | 0.460 | 0.459 |
| Independent | 0.515 | 0.501 | 0.477 | 0.465 | | | 0.454 | | 0.448 |
| Ours (ESA) 3.3.1 | **0.503** | 0.487 | 0.467 | 0.461 | 0.457 | 0.451 | 0.451 | 0.447 | 0.447 |
| Ours (HOA) 3.3.2 | **0.503** | **0.461** | 0.455 | 0.451 | 0.449 | 0.445 | 0.444 | 0.445 | 0.442 |
| Ours Optimal | **0.503** | **0.461** | **0.452** | **0.446** | **0.442** | **0.439** | **0.436** | **0.436** | **0.435** |

Table 7: The total test set loss on all five tasks for each method under each inference time budget. Lower is better. The data is the same as in Figures 3 and 2.

|        | SemSeg  | Depth  | Normals | Keypoints | Edges   |
|--------|---------|--------|---------|-----------|---------|
| S      | 0.08039 | –      | –       | –         | –       |
| D      | –       | 0.1695 | –       | –         | –       |
| N      | –       | –      | 0.08591 | –         | –       |
| K      | –       | –      | –       | 0.0895    | –       |
| E      | –       | –      | –       | –         | 0.02783 |
| SD     | 0.07858 | 0.1833 | –       | –         | –       |
| SN     | 0.074   | –      | 0.0997  | –         | –       |
| SK     | 0.07722 | –      | –       | 0.09718   | –       |
| SE     | 0.07897 | –      | –       | –         | 0.04462 |
| DN     | –       | 0.1695 | 0.09275 | –         | –       |
| DK     | –       | 0.1706 | –       | 0.09318   | –       |
| DE     | –       | 0.1748 | –       | –         | 0.03192 |
| NK     | –       | –      | 0.08968 | 0.09181   | –       |
| NE     | –       | –      | 0.09358 | –         | 0.02908 |
| KE     | –       | –      | –       | 0.09185   | 0.03488 |
| SDN    | 0.07498 | 0.1698 | 0.09575 | –         | –       |
| SDK    | 0.07699 | 0.1782 | –       | 0.09704   | –       |
| SDE    | 0.07893 | 0.1863 | –       | –         | 0.04559 |
| SNK    | 0.0722  | –      | 0.09919 | 0.0961    | –       |
| SNE    | 0.07222 | –      | 0.0982  | –         | 0.03689 |
| SKE    | 0.0766  | –      | –       | 0.09342   | 0.03508 |
| DNK    | –       | 0.1654 | 0.09358 | 0.09253   | –       |
| DNE    | –       | 0.1708 | 0.09396 | –         | 0.03286 |
| DKE    | –       | 0.1793 | –       | 0.09073   | 0.02937 |
| NKE    | –       | –      | 0.09626 | 0.09024   | 0.02609 |
| SDNK   | 0.07762 | 0.1822 | 0.09869 | 0.1015    | –       |
| SDNE   | 0.07576 | 0.1735 | 0.09718 | –         | 0.04513 |
| SDKE   | 0.0795  | 0.1797 | –       | 0.09272   | 0.04141 |
| SNKE   | 0.07369 | –      | 0.09944 | 0.09697   | 0.03312 |
| DNKE   | –       | 0.1708 | 0.09392 | 0.09334   | 0.02803 |
| SDNKE  | 0.07854 | 0.1864 | 0.1     | 0.09814   | 0.04453 |

Table 8: **The validation set performance of our 31 networks on each task that they solve.** Tasks are named to contain a letter for each task that they solve. *S: Semantic Segmentation, D: Depth Estimation, N: Surface Normal Prediction, K: Keypoint Detection, E: Edge Detection.*

| | SemSeg | Depth | Normals | Keypoints | Edges |
|---|---|---|---|---|---|
| S | 0.07662 | – | – | – | – |
| D | – | 0.1696 | – | – | – |
| N | – | – | 0.08555 | – | – |
| K | – | – | – | 0.08847 | – |
| E | – | – | – | – | 0.0275 |
| SD | 0.07419 | 0.1831 | – | – | – |
| SN | 0.07084 | – | 0.0994 | – | – |
| SK | 0.07369 | – | – | 0.09601 | – |
| SE | 0.07504 | – | – | – | 0.044 |
| DN | – | 0.1694 | 0.09249 | – | – |
| DK | – | 0.1713 | – | 0.08882 | – |
| DE | – | 0.1753 | – | – | 0.03145 |
| NK | – | – | 0.08934 | 0.09077 | – |
| NE | – | – | 0.09327 | – | 0.02865 |
| KE | – | – | – | 0.09077 | 0.0344 |
| SDN | 0.07193 | 0.17 | 0.09544 | – | – |
| SDK | 0.07311 | 0.1785 | – | 0.09591 | – |
| SDE | 0.07617 | 0.1865 | – | – | 0.04474 |
| SNK | 0.06933 | – | 0.09966 | 0.09302 | – |
| SNE | 0.06859 | – | 0.09796 | – | 0.03625 |
| SKE | 0.07323 | – | – | 0.09232 | 0.03463 |
| DNK | – | 0.1658 | 0.09318 | 0.09143 | – |
| DNE | – | 0.1706 | 0.09362 | – | 0.03239 |
| DKE | – | 0.1795 | – | 0.08968 | 0.02887 |
| NKE | – | – | 0.09596 | 0.08921 | 0.02566 |
| SDNK | 0.07338 | 0.1826 | 0.09836 | 0.1003 | – |
| SDNE | 0.07249 | 0.1739 | 0.09689 | – | 0.04441 |
| SDKE | 0.07634 | 0.1801 | – | 0.09157 | 0.04097 |
| SNKE | 0.07111 | – | 0.09941 | 0.09464 | 0.03328 |
| DNKE | – | 0.1704 | 0.09356 | 0.09226 | 0.02768 |
| SDNKE | 0.07603 | 0.186 | 0.09976 | 0.09704 | 0.04395 |

Table 9: **The test set performance of our 31 networks on each task that they solve.**

