# OpenReview forum: "Which Tasks Should Be Learned Together in Multi-task Learning?"
_ICLR.cc/2020/Conference — Reject_

### Official Review · AnonReviewer2 · 2019-10-23
**Official Blind Review #2**

**Rating:** 6

**Review:**

This submission studies how to group tasks to train together to find a better runtime-accuracy trade-off with a single but large multi-task neural model. The authors perform an extensive empirical study in the Taskonomy dataset. I like Section 6.1 in particular, which shows the difference between multi-task learning and transfer learning.

My main concern is that the competition between different tasks may stem from the limited capacity of the model during training. It might be possible that with enough parameters, the competing tasks become compatible. If I were the authors, I would first train a bigger model on multiple tasks and then distill it into a smaller one, which does not increase inference time.

Furthermore, the paper has more than 8 pages.

**Experience Assessment:**

I have read many papers in this area.

**Review Assessment: Checking Correctness Of Derivations And Theory:**

N/A

**Review Assessment: Checking Correctness Of Experiments:**

I assessed the sensibility of the experiments.

**Review Assessment: Thoroughness In Paper Reading:**

I read the paper at least twice and used my best judgement in assessing the paper.

---

> ### Author Response · Authors · 2019-11-15
> **Response to Review #2**
>
> Thank you for your review and your appreciation of the section comparing transfer learning and multi-task learning. We value your comments and will be updating the final version according to the discussion below.
>
> The data in Figure 3 shows the performance of all-in-one networks and individual networks as the number of parameters grows. The trend seems to be that as the number of parameters increases, training individually actually beats all-in-one networks by wider and wider margins. Thus, competition actually seems to increase as the number of parameters grows. Nevertheless, a distillation approach seems interesting and MTL performance might be improved by distilling from very large networks. We are not aware of any work exploring this technique, but it might be an interesting topic for a separate study of its own.
>
> If distillation does work for MTL, it could be combined with our technique, possibly leading to even better performance.
>
> We've edited the paper to exactly 8 pages by removing unnecessary section headings and making minor wording changes.

---

### Official Review · AnonReviewer1 · 2019-10-24
**Official Blind Review #1**

**Rating:** 6

**Review:**

This paper focuses on how to partition a bunch of tasks in several groups and then it use multi-task learning to improve the performance.  The paper makes an observation that multi-task relationships are not entirely correlated to transfer relationships and proposes a computational framework to optimize the assignment of tasks to network under a given computational budget constraint. It experiments on different combinations of the tasks and uses two heuristics to reduce the training overheads, early stopping approximation and higher order approximation.

Please see the detained comments as follows:
1. The experiments are based on the assumptions that the network structures (how parameters are shared across tasks) are fixed. From my perspective, understanding how to optimize the parameters sharing across two tasks should be the first step to study how to optimally combine the training of tasks. Otherwise, different parameter sharing structures across tasks may lead to different conclusions.

2. It requires optimization of the article structure. E.g., algorithm 1 is important and should be in the main context.

3. It is also related to neural architecture search and it requires some discussions.

4. The paper is over-length.

5. A lot of typos.

Nits:
Page 1: vide versa -> vice versa

Page 3: two networsk -> two networks

Page 6: budge ?? 1.5

Page 7: overlap between lines (under figure 3)

Page 8: half segmented sentence in section 6.


**Experience Assessment:**

I have published in this field for several years.

**Review Assessment: Checking Correctness Of Derivations And Theory:**

I did not assess the derivations or theory.

**Review Assessment: Checking Correctness Of Experiments:**

I carefully checked the experiments.

**Review Assessment: Thoroughness In Paper Reading:**

I read the paper at least twice and used my best judgement in assessing the paper.

---

> ### Author Response · Authors · 2019-11-15
> **Response to Review #1**
>
> Thank you for your review. We appreciate your comments.
>
> 1. How to best share parameters:
> We agree that finding the best way to share parameters is an open problem and a promising direction for future work. However, we believe that a good technique for determining which parameters to share could be used in conjunction with our techniques to yield further performance improvements.
>
> In our work, we chose the [encoder->multiple decoder] structure because a number of recent and high profile MTL works use it (Kokkinos 2016 (Ubernet), Chen et al 2018 (Gradnorm), Sener et al 2018, Kendall et al. (2018) and many more). Furthermore, we considered two additional sharing schemes, [U-Net with multiple output channels] and [encoder->single decoder with multiple output channels]. Both were found to be inferior to [encoder->multiple decoder]. Experimental numbers below for 1-SNT networks trained on all tasks jointly:
>
>                                                           total_loss (lower is better)
> U-Net                                                               0.414                               14.05% Worse
> encoder->single decoder                             0.377                               3.85% Worse
> encoder->multiple decoders (ours)           0.363
> *note that these numbers cannot be compared with those in the paper due to differences in the loss function.
>
> This is far from a comprehensive search of parameter sharing possibilities, but out of the techniques commonly used in the literature, the one we chose performs the best.
>
> 2. Algorithm 1.
> Thanks for the suggestion. We believe that the particular algorithm chosen for our optimization problem is a detail because many algorithms would work equally well, but we can move Algorithm 1 to the main text if other reviewers agree.
>
> 3. Neural Architecture Search section.
> We've added a neural architecture search section in the related work.
>
> 4. Paper length.
> We've edited the paper to exactly 8 pages by removing unnecessary section headings and making minor wording changes.
>
> 5. Typos.
> Thank you for these edits. We've gone through the entire paper and fixed typos and grammatical errors.

---

### Official Review · AnonReviewer3 · 2019-10-25
**Official Blind Review #3**

**Rating:** 6

**Review:**

This paper works on the problem if training a set of networks to solve a set of tasks. The authors try to discover an optimal task split into the networks so that the test performances are maximized given a fixed testing resource budget. By default, this requires searching over the entire task combination space and is too slow. The authors propose two strategies for fast approximating the enumerative search. Experiments show their searched combinations give better performance in the fixed-budget testing setting than several alternatives.

+ This paper works on a new and interesting problem. Training more than one networks for a few tasks is definitely a valid idea in real applications and related to broad research fields.
+ The baseline setup is comprehensive. The difference between optimal, random, and worst clearly shows this problem worths effort for research.
- I believe this problem setup requires a larger task set. In the paper the authors manually picked 5 tasks. It seems straightforward for a human to manually group them together: segmentation and edge/ surface + depth/ keypoint for 3 networks. It is unclear how better a network can do than a human in one minute or should we expect learning the task split is better than manual design.
- Both technical contributions in Section 3.3 look straightforward. Given the good performance in Figure3, it is fine.
- I am confused by the comparison to Sener and Koltun. How do you change the inference budget for them? If it is changing the number of channels for a single network, I believe it can be improved more.

Overall I believe this is a good paper to open interesting research direction with solid baselines. I am happy to accept this paper and see more exciting future works in this direction.

**Experience Assessment:**

I have read many papers in this area.

**Review Assessment: Checking Correctness Of Derivations And Theory:**

I assessed the sensibility of the derivations and theory.

**Review Assessment: Checking Correctness Of Experiments:**

I assessed the sensibility of the experiments.

**Review Assessment: Thoroughness In Paper Reading:**

I read the paper thoroughly.

---

> ### Author Response · Authors · 2019-11-15
> **Response to Review #3**
>
> Thank you for your review and finding the submission “a good paper to open interesting research direction with solid baselines”. We appreciate your comments and will be updating the final version according to the discussion below.
>
> Regarding manually designed groupings, Figure 5 shows that there seems to be a negative correlation between multi-task affinity and the more intuitive transfer learning affinity of Taskonomy. This makes it seem unlikely that a human would be able to do a good job of engineering task groupings without extensive experience. Furthermore, humans are unlikely to leverage the full potential of auxiliary tasks (i.e., tasks that help other tasks when trained with them but have better performance when trained separately).
>
> As an example, using the data in Table 9, we can see how well the grouping you suggest performs. We can also evaluate an alternative grouping supported by intuition (e.g., group the two 3D tasks together and the two 2D tasks together). The table below shows how each of those groupings perform versus the computationally found optimal grouping:
>
>         Grouping                                           Total Loss (lower is better)
> optimal 3 grouping (SDn,N,nKE)                       0.44235
> Your suggested grouping (S, NE, DK)              0.45866
> 2D/3D grouping (S, DN, KE)                               0.46368
>
> We see that the two human designed groupings underperform the optimal grouping found computationally. In addition, the optimal task grouping is likely architecture/data-dependent at least to a certain extent, which supports developing computational methods for finding them, in contrast to fixed human intuitions.
>
> As for the comparison to Sener and Koltun (and single traditional network), the reason we chose to make the network wider rather than deeper is that wider networks can emulate multiple networks, but it is not clear that deeper ones can. Nevertheless it would be interesting to see how well deeper (or some combination of deeper and wider) networks perform.  We will have this comparison for the camera ready, but we expect they will perform similarly to the wider networks we use.

---

### Author Response · Authors · 2019-11-15
**Thanks**

We thank the reviewers and meta-reviewers for their consideration.

We've edited the paper to exactly 8 pages by removing unnecessary section headings and making minor wording changes.

---

### Decision · Program_Chairs · 2019-12-19

**Decision:**

Reject

**Comment:**

An approach to make multi-task learning is presented, based on the idea of assigning tasks through the concepts of cooperation and competition.

The main idea is well-motivated and explained well. The experiments demonstrate that the method is promising. However, there are a few  concerns regarding fundamental aspects, such as: how are the decisions affected by the number of parameters? Could ad-hoc algorithms with human in the loop provide the same benefit, when the task-set is small? More importantly, identifying task groups for multi-task learning is an idea presented in prior work, e.g. [1,2,3]. This important body of prior work is not discussed at all in this paper.

[1] Han and Zhang. "Learning multi-level task groups in multi-task learning"
[2] Bonilla et al. "Multi-task Gaussian process prediction"
[3] Zhang and Yang. "A Survey on Multi-Task Learning"